# Role of RhoGEFs or RhoGAPs in Pyk2-Mediated RhoA Activation in Depolarization-Induced Contraction of Rat Caudal Arterial Smooth Muscle

**DOI:** 10.3390/ijms26178676

**Published:** 2025-09-05

**Authors:** Kazuki Aida, Mitsuo Mita, Reiko Ishii-Nozawa

**Affiliations:** 1Department of Pharmacology, Meiji Pharmaceutical University, 2-522-1 Noshio, Kiyose 204-8588, Tokyo, Japan; reiko-in@my-pharm.ac.jp; 2Professor Emeritus, Meiji Pharmaceutical University, 2-522-1 Noshio, Kiyose 204-8588, Tokyo, Japan; mitsuom@my-pharm.ac.jp

**Keywords:** proline-rich tyrosine kinase 2, RhoA, Rho guanine nucleotide exchange factor, Rho GTPase-activating protein

## Abstract

It has previously been reported that the RhoA/Rho-associated kinase (ROCK) pathway is involved in depolarization-induced contraction triggered by high [K^+^] stimulation in rat caudal arterial smooth muscle. Furthermore, we reported that activation of the upstream Ca^2+^-dependent proline-rich tyrosine kinase 2 (Pyk2) leads to phosphorylation of myosin targeting subunit of myosin light chain phosphatase (MYPT1) and 20 kDa myosin light chain (LC_20_). These findings suggest that Rho guanine nucleotide exchange factors (RhoGEFs) or Rho GTPase-activating proteins (RhoGAPs) may mediate RhoA activation downstream of Pyk2, thereby contributing to depolarization-induced contraction. However, it remains unclear whether Pyk2 directly interacts with RhoGEFs or RhoGAPs. In this study, we investigated the interaction between Pyk2 and RhoGEFs or RhoGAPs during depolarization stimulation of rat caudal arterial smooth muscle. We examined the interaction between Pyk2 and RhoGEFs or RhoGAPs, which previously were identified in smooth muscle, specifically in rat caudal arterial smooth muscle, in response to 60 mM K^+^ stimulation by immunoprecipitation analysis. ArhGEF11, ArhGEF12, phosphorylated ArhGAP42 at Tyr792 (pTyr792-ArhGAP42) and phosphorylated ArhGAP42 at Tyr376 (pTyr376-ArhGAP42) co-immunoprecipitated with Pyk2. The co-immunoprecipitation of pTyr792-ArhGAP42, but not pTyr376-ArhGAP42, with Pyk2 was inhibited by a Pyk2 inhibitor, sodium salicylate. Furthermore, 60 mM K^+^ stimulation increased ArhGAP42 phosphorylation at Tyr792, which was also suppressed by sodium salicylate. These findings indicate that Pyk2-mediated phosphorylation of ArhGAP42 at Tyr792 may play a role in depolarization-induced contraction of rat caudal arterial smooth muscle.

## 1. Introduction

The contraction of vascular smooth muscle is regulated by the balance between phosphorylation of the 20 kDa light chain of myosin (LC_20_), catalyzed by Ca^2+^/calmodulin (CaM)-dependent myosin light chain kinase (MLCK), and dephosphorylation, catalyzed by myosin light chain phosphatase (MLCP) [1,2,3].

In recent years, increasing attention has been paid to the molecular mechanisms of force regulation that are independent of changes in cytosolic free Ca^2+^ concentration ([Ca^2+^]_i_), so-called Ca^2+^ sensitization [3]. Ca^2+^ sensitization is primarily regulated through the inhibition of MLCP, leading to enhanced LC_20_ phosphorylation [3].

RhoA plays a central role in Ca^2+^ sensitization. The active form, RhoA-GTP, activates Rho-associated kinase (ROCK), and activated ROCK phosphorylates the myosin-targeting subunit of MLCP (MYPT1). Phosphorylation of MYPT1 leads to a decrease in phosphatase activity, resulting in increased phosphorylation of LC_20_ and enhanced contraction of smooth muscle [1,3]. Thus, activation of the RhoA/ROCK pathway serves as a major downstream signaling cascade for receptor- and G protein-mediated Ca^2+^ sensitization [3,4]. Moreover, RhoA/ROCK pathways in vascular smooth muscles plays an important role in the pathogenesis of hypertension, coronary artery spasm and contributes to angina, myocardial infarction, and sudden death [5,6,7].

Rho proteins, which are small GTPases, function as molecular switches by adopting different conformations depending on whether they are bound to GDP or GTP. Rho activity is promoted by guanine nucleotide exchange factors (RhoGEFs), which catalyze the exchange of GDP for GTP, and restored by Rho GTPase-activating proteins (RhoGAPs), which stimulate the hydrolysis of GTP to GDP, in response to various extracellular stimuli [8], ultimately regulating many cellular responses, such as cell morphology, growth, and motility [9]. In cardiovascular diseases including hypertension, RhoGEFs and RhoGAPs are believed to be key molecules involved in the hyperactivation of RhoA [10,11,12,13]. Currently, ~80 RhoGEFs and ~70 RhoGAPs have been identified as regulators of Rho GTPases, and more than 30 effectors have been characterized. Among these, several RhoGEFs and RhoGAPs, including ArhGEF1 (p115-RhoGEF) [14,15,16], ArhGEF2 (GEF-H1) [17], ArhGEF11 (PDZ-RhoGEF) [18,19], ArhGEF12 (LARG) [20,21,22,23,24], ArhGEF18 (p114-RhoGEF) [25,26], ArhGEF25 (p63-RhoGEF) [27,28], ArhGAP35 (p190-RhoGAP) [29,30], ArhGAP42 (GRAF3) [12,31], have been reported to play roles in the regulation of blood pressure [11,13]. However, our understanding of the expression, activity, and regulation of RhoGEFs or RhoGAPs in vascular smooth muscle cells remains limited.

Electromechanical coupling operates through changes in membrane potential, which influence [Ca^2+^]_i_. High extracellular [K^+^] stimulation induces membrane depolarization, which opens voltage-dependent Ca^2+^ channels and leads to Ca^2+^ influx. This results in increased [Ca^2+^]_i_, Ca^2+^/CaM binding, MLCK activation, LC_20_ phosphorylation, and contraction [2,3]. We previously demonstrated that 60 mM K^+^-induced membrane depolarization causes a rapid contraction, followed by a sustained steady-state contraction greater than the resting tone in endothelium-denuded rat caudal arterial smooth muscle [32]. The phasic rapid contraction is due to an increase in [Ca^2+^]_i_ and LC_20_ phosphorylation catalyzed by MLCK, while the sustained contraction involves activation of the RhoA/ROCK pathway and inhibition of MLCP [32]. Furthermore, the sustained contraction, RhoA activation, and increased MYPT1 phosphorylation induced by high [K^+^] and ionomycin were suppressed by the Pyk2 inhibitors, sodium salicylate and PF-431396, suggesting that Ca^2+^-dependent proline-rich tyrosine kinase 2 (Pyk2, also known as FAK2, CAKβ, or RAFTK) functions upstream of RhoA activation in response to membrane depolarization in endothelium-denuded rat caudal arterial smooth muscle, thereby contributing to MLCP inhibition and the resulting sustained contraction [33,34,35]. In addition, phosphorylation of Pyk2 at Tyr402 was increased in caudal arterial smooth muscle following treatment with high [K^+^] and ionomycin, and this effect was inhibited by the Pyk2 inhibitors sodium salicylate and PF-431396 [33,34,35,36]. However, the mechanism by which Pyk2, which is involved in membrane depolarization-induced contraction, activates the RhoA/ROCK pathway remains unclear.

Pyk2, which may function upstream of RhoA/ROCK, is a non-receptor, Ca^2+^-dependent protein tyrosine kinase regulated by various extracellular signals and elevations in [Ca^2+^]_i_ [37,38,39]. Although the precise mechanism by which Ca^2+^ activates Pyk2 remains unclear, the Ca^2+^/calmodulin (CaM) complex is known to bind to the FERM domain of Pyk2. This domain is thought to mediate Ca^2+^/CaM-dependent Pyk2 homodimerization and trans-autophosphorylation at Tyr402, which facilitates Src recruitment. Subsequently, Src phosphorylates Pyk2 at Tyr579 and Tyr580 within the activation loop of the kinase domain, leading to maximal Pyk2 activation [40]. However, the involvement of RhoGEFs or RhoGAPs in the Pyk2-mediated Ca^2+^-dependent RhoA/ROCK pathway activation is unclear. One possible explanation is that RhoGEFs or RhoGAPs, which are known to undergo tyrosine phosphorylation [41,42,43], may be modified by activated Pyk2 in response to a sustained increase in [Ca^2+^]_i_ following membrane depolarization.

In this study, we investigated the interaction between Pyk2 and RhoGEFs or RhoGAPs in the membrane depolarization-induced sustained contraction of rat caudal arterial smooth muscle using co-immunoprecipitation and Western blotting, and demonstrated that specific RhoGEFs and RhoGAPs are involved in the Ca^2+^-dependent RhoA/ROCK pathway mediated by Pyk2.

## 2. Results

Among the RhoGEFs and RhoGAPs that may be involved in vascular smooth muscle contraction [11,12,13,27], six RhoGEFs (ArhGEF1, ArhGEF2, ArhGEF11, ArhGEF12, ArhGEF18, and ArhGEF25) and two RhoGAPs (ArhGAP35 and ArhGAP42) were analyzed by qRT-PCR. Transcripts for each RhoGEF and RhoGAP was detected in the vascular smooth muscle tissue of the rat caudal artery, and the presence of six RhoGEFs and two RhoGAPs was confirmed, but their expression levels were not compared using absolute quantification of the PCR products (Figure 1).

We used co-immunoprecipitation and Western blotting to examine the protein expression of each RhoGEF and RhoGAP, as well as their interaction with phosphorylated Pyk2 at Tyr402 (pTyr402-Pyk2) during the contraction induced by depolarization with 60 mM K^+^ for 5 min. The reason for using 5 min of stimulation in this experiment is that previous research has shown that Pyk2 phosphorylation reaches its maximum within 5 min of stimulation [34]. In addition, a Pyk2 inhibitor sodium salicylate was used at a concentration of 10 mM, which effectively inhibited sustained contractions without affecting the phasic contraction induced by 60 mM K^+^, in contrast to another Pyk2 inhibitor, PF431396 [34,36]. The expression of each ArhGEF and of actin was analyzed using tissue homogenate samples (bottom two panels of Figure 2(Aa–f,B)). Protein expression was confirmed for all examined RhoGEFs, with ArhGEF18 and ArhGEF25 tending to be more highly expressed (Figure 2(Ae,Af,B)). Moreover, co-immunoprecipitation with pTyr402-Pyk2 revealed bands for ArhGEF11 and ArhGEF12 (Figure 2(Ac,Ad,C)). However, no significant differences were observed in response to 60 mM K^+^ stimulation, either in the absence or presence of 10 mM sodium salicylate (Figure 2(Ac,Ad,C)).

The expression of each ArhGAP and of actin was analyzed using tissue homogenate samples (bottom two panels of Figure 3(Aa–c,B)). Because no commercially available anti-ArhGAP42 antibodies cross-react with rat proteins, and the antibodies we produced exhibited poor binding, we resorted to using anti-phosphorylated (Tyr792) ArhGAP42 antibody and anti-phosphorylated (Tyr376) ArhGAP42 antibody. The expression of ArhGAP35 and ArhGAP42 was verified (Figure 3(Aa–c,B)). Assessment of ArhGAP42 phosphorylation at Tyr792 and Tyr376 in the homogenate fraction demonstrated that ArhGAP42 phosphorylation at Tyr792 was significantly increased by 60 mM K^+^ stimulation and returned to the unstimulated level in the presence of 10 mM sodium salicylate (Figure 3(Ca)). Conversely, ArhGAP42 phosphorylation at Tyr376 was neither increased by 60 mM K^+^ stimulation nor affected by 10 mM sodium salicylate (Figure 3(Da)). Furthermore, co-immunoprecipitation with Pyk2 revealed that ArhGAP42 phosphorylation at Tyr792 and Tyr376 was associated with Pyk2 (Figure 3(Ab,Ac,Cb,Db)). The interaction between Pyk2 and ArhGAP42 phosphorylated at Tyr792 (pTyr792-ArhGAP42) showed an increasing trend upon 60 mM K^+^ stimulation, although this change did not reach statistical significance (Figure 3(Ab,Cb)). However, the association was significantly attenuated under 60 mM K^+^ stimulation in the presence of 10 mM sodium salicylate (Figure 3(Ab,Cb)). In contrast, the co-immunoprecipitation of ArhGAP42 phosphorylated at Tyr376 (pTyr376-ArhGAP42) with Pyk2 remained unchanged upon 60 mM K^+^ stimulation, both in the presence and absence of 10 mM sodium salicylate (Figure 3(Ac,Db)).

## 3. Discussion

We previously reported that Pyk2, activated by a rise in [Ca^2+^]_i_ resulting from depolarization-induced Ca^2+^ influx, is involved in activation of the RhoA/ROCK pathway, leading to the phosphorylation of MYPT1, during the sustained contraction of rat caudal arterial smooth muscle induced by 60 mM K^+^ stimulation [32,33,34,35]. However, the molecular mechanism by which activated Pyk2 promotes RhoA/ROCK activation remains to be elucidated. In the present study, we investigated whether Pyk2 regulates RhoA activity through modulation of RhoGEFs or RhoGAPs.

RhoA activation is regulated by RhoGEFs and RhoGAPs, which promote and inhibit its activity, respectively [9]. Among the various RhoGEFs and RhoGAPs, ArhGEF1 has been reported to participate in the angiotensin II-induced RhoA activation mechanism and is phosphorylated by JAK2 [14]. ArhGEF2 is involved in the activation of the RhoA/ROCK pathway in pulmonary artery smooth muscle and contributes to smooth muscle proliferation [17]. ArhGEF11 has been shown to co-immunoprecipitate with RhoA in mesenteric arteries stimulated with angiotensin II [18]. ArhGEF12 enhanced RhoA activation and increased MYPT1 phosphorylation in vascular smooth muscle cells in response to angiotensin II [20]. ArhGEF18 activated RhoA via Gβγ signaling in cultured cells [25], and ArhGEF25 promoted RhoA activation through Gαq/11 in mouse portal vein upon stimulation with angiotensin II or endothelin-1 [27]. ArhGAP35 facilitated RhoA activation through dephosphorylation mediated by Src homology 2 domain-containing phosphatase 2 (SHP2) in rat aortic smooth muscle cells [29]. In contrast, ArhGAP42 knockdown led to increased RhoA activation and myosin light chain phosphorylation in rat aortic smooth muscle cells [12]. Therefore, we selected these RhoGEFs and RhoGAPs as potential target proteins of Pyk2 because of their involvement in RhoA/ROCK activation in blood vessels [11,13]. In this study, expression of both mRNA and protein of these RhoGEFs and RhoGAPs was confirmed in rat caudal arterial smooth muscle (Figure 1, Figure 2 and Figure 3). However, in this study, no commercially available anti-ArhGAP42 antibodies cross-reacted with rat proteins, and the antibodies we generated showed poor binding; therefore, we had no choice but to use an anti-phosphorylated (Tyr792) ArhGAP42 antibody and an anti-phosphorylated (Tyr376) ArhGAP42 antibody. Nonetheless, since both phosphorylated-specific antibodies successfully detected the protein, we believe this does not pose any problems. Therefore, the protein expression level of ArhGAP42 could not be compared with those of other ArhGEFs and ArhGAP35. In particular, the protein levels of ArhGEF18 and ArhGEF25 were relatively high. It has been reported that all three members of the RGS-containing Rho-GEF subfamily, which are ArhGEF1 (p115-RhoGEF), ArhGEF11 (PDZ-RhoGEF), and ArhGEF12 (LARG), are expressed in arteries, with ArhGEF11 showing the highest expression at both mRNA and protein levels in the rat aorta and mesenteric artery [16,18,44]. In contrast, in the mouse aorta, ArhGEF12 has been identified as the predominantly expressed RGS-Rho-GEF, whereas ArhGEF11 shows the lowest expression [22]. These differences between previous findings and our results are likely attributable to species- and tissue-specific differences.

Pyk2 is a non-receptor tyrosine kinase regulated by various extracellular signals, activated by elevated [Ca^2+^]_i_ [39], and reported to be inhibited by high concentrations of sodium salicylate [45,46]. In our previous report, we demonstrated that in endothelium-denuded rat caudal arterial smooth muscle, sodium salicylate suppressed the sustained, but not the phasic, contraction, RhoA activation (assessed by RhoA translocation to the plasma membrane), and the increase in MYPT1 phosphorylation induced by high [K^+^] or ionomycin [33,34,35]. Furthermore, although Pyk2 is activated via autophosphorylation at Tyr402 [40], high [K^+^] and ionomycin stimulation were reported to increase Pyk2 phosphorylation at this site in caudal arterial smooth muscle, and this phosphorylation was inhibited by sodium salicylate [33,34,35,36]. PF-431396, a well-known potent Pyk2/focal adhesion kinase (FAK) inhibitor [47], was previously shown by us to suppress high K^+^-induced contraction, Pyk2 autophosphorylation and LC_20_ phosphorylation [34]. However, unlike sodium salicylate, PF-431396 suppressed not only the sustained but also the phasic component of 60 mM K^+^-induced contraction [34,36], suggesting potential non-specific effects beyond Pyk2 inhibition [47]. Accordingly, sodium salicylate, considered a more Pyk2-selective inhibitor, was used in the present study. In this study, four proteins—ArhGEF11, AhrGEF12 (Figure 2), pTyr792-ArhGAP42, and pTyr376-ArhGAP42 (Figure 3)—were co-immunoprecipitated with phosphorylated Pyk2 at Tyr402 in rat caudal arterial smooth muscle. Among them, only the co-immunoprecipitation of pTyr792-ArhGAP42 with phosphorylated Pyk2 was increased by 60 mM K^+^ stimulation, and this increase was inhibited by sodium salicylate (Figure 3). A previous report has shown that ArhGEF11 co-immunoprecipitates with Pyk2 via TROY in HEK293 cells and activate RhoA, whereas ArhGEF12 does not co-immunoprecipitate with Pyk2 [48]. It has also been reported that angiotensin II stimulation induces Pyk2-mediated phosphorylation of ArhGEF11, leading to RhoA activation in rat aortic smooth muscle cells [19]. We previously reported that the RhoGEF inhibitor Y16 [49], a cell-permeable compound that targets the junction of the DH-PH domains of RhoGEFs with high affinity (K_d_ = 65 nM) and effectively prevents the interaction between RhoA and RhoGEFs such as ArhGEF1 (p115-RhoGEF), ArhGEF11 (PDZ-RhoGEF) and ArhGEF12 (LARG) [49], had no detectable effect on contractile response induced by depolarization with 60 mM K^+^ stimulation [50]. The present results, showing no significant difference in immunoprecipitate binding between 60 mM K^+^ stimulation and inhibition of Pyk2 activity by sodium salicylate, suggest that Pyk2-mediated activation of ArhGEF11 and ArhGEF12 plays only a minor role in RhoA activation during sustained contraction induced by 60 mM K^+^ in the absence of receptor stimulation. Nevertheless, a potential contribution of these RhoGEFs at certain time points during K^+^-induced contraction cannot be fully excluded, in contrast to receptor-mediated stimulation such as that induced angiotensin II. On the other hand, ArhGAP42 is a RhoGAP specifically expressed in vascular smooth muscle, and its knockdown has been reported to increase blood pressure and enhance RhoA activation [12]. It has also been reported that ArhGAP42 acts as a blood volume-sensitive rheostat that suppresses excessive RhoA activation, thereby contributing to the regulation of blood pressure [31]. To our knowledge, this is the first report to demonstrate that Pyk2 and ArhGAP42 co-immunoprecipitate in intact vascular smooth muscle, and that this interaction is inhibited by sodium salicylate (Figure 3). Furthermore, in the homogenate fraction, 60 mM K^+^ stimulation increased ArhGAP42 phosphorylation at Tyr792 compared to unstimulated condition (Figure 3). This increase was inhibited by sodium salicylate, restoring the phosphorylation level to that of the unstimulated state (Figure 3). On the other hand, no increase in ArhGAP42 phosphorylation at Tyr376 was observed with 60 mM K^+^ stimulation (Figure 3). Regarding ArhGAP42 phosphorylation at Tyr376, Luo et al. reported that in mouse embryonic fibroblasts, the tyrosine kinase Src phosphorylates ArhGAP42 at Tyr376, relieving the inhibitory effect of the BAR domain on GAP activity and thereby suppressing RhoA activation [51]. In our study, stimulation with 60 mM K^+^ did not induce any changes in ArhGAP42 phosphorylation at Tyr376, suggesting that RhoA inactivation through ArhGAP42 phosphorylation at Tyr376 likely did not occur in rat caudal arterial smooth muscle. Thus, these findings suggest that Pyk2-mediated phosphorylation of ArhGAP42 at Tyr792, but not at Tyr376, may contribute to RhoA activation during the sustained contraction induced by 60 mM K^+^-mediated depolarization. However, a key limitation of this study is that the functional consequences of Tyr792 phosphorylation on ArhGAP42 activity remains unresolved. To the best of our knowledge, there have been no reports on functional changes in the GAP activity of ArhGAP42 resulting from Tyr792 phosphorylation. Wolf et al. reported that ArhGAP35 (p190-RhoGAP) is phosphorylated by Fyn, which promotes its association with p120RasGAP, enhances GAP activity, and subsequently restrains RhoA [52]. On the other hand, Lee et al. demonstrated that ArhGAP26 (GRAF1), a protein closely related to ArhGAP42, co-immunoprecipitates with Pyk2 in cultured mouse neurons, and that Pyk2-mediated phosphorylation suppresses its GAP activity [53]. Collectively, these findings underscore the intricate regulatory role of tyrosine phosphorylation in modulating GAP function, and highlight the critical need for further investigation to elucidate how Tyr792 phosphorylation specifically influences ArhGAP42 activity. Moreover, the faint detection of RhoGEFs and RhoGAPs in the tissue-derived homogenate fraction likely reflects their low endogenous expression levels (Figure 2 and Figure 3). Therefore, we consider that future studies should use cultured cells with elevated RhoGEF and RhoGAP expression to obtain clearer and more mechanistic insights.

## 4. Materials and Methods

### 4.1. Materials

Prazosin (1-[4-amino-6, 7-dimethoxy-2-quinazolinyl]-4-[2-furanylcarbonyl]-piperazine), DL-propranolol (1-[isopropylamino]-3-[1-naphthyloxy]-2-propanol) were purchased from Sigma-Aldrich (St. Louis, MO, USA). NaCl, KCl, CaCl_2_, MgCl_2_, glucose and sodium salicylate from Wako Pure Chemical Industries (Wako) (Osaka, Japan) and 2-[4-(2-Hydroxyethyl)-1-piperazinyl]ethanesulfonic acid (Hepes) from Dojindo Laboratories (Kumamoto, Japan). All other chemicals were of reagent grade. Stock solutions were prepared in water for prazosin and propranolol.

### 4.2. Preparation of Smooth Muscle Strips

Male Sprague-Dawley rats (400–550 g) were anesthetized with 5% isoflurane and euthanized by carbon dioxide as approved by the Institutional Ethics Committee for Animal Research at Meiji Pharmaceutical University and conforming to Guidelines for Proper Conduct of Animal Experiments in Japan. Caudal arteries were removed and helical strips [0.5 mm × (6–7) mm] were cut in Hepes-Tyrode (H-T) solution (137 mM NaCl, 2.7 mM KCl, 1.8 mM CaCl_2_, 1 mM MgCl_2_, 5.6 mM glucose, 10 mM Hepes, pH 7.4). The endothelium was mechanically removed from all segments by gentle rubbing. 60 mM K^+^-induced contraction was carried out in the presence of 1 µM prazosin and 0.1 µM propranolol to block the α_1_- and β-adrenergic effects of noradrenaline, which is released from nerve terminals [32]. All buffers were pre-oxygenated with 100% O_2_ at room temperature.

### 4.3. qRT-PCR

Based on sequence homology and bibliographic data, we identified six mRNA sequences of rat RhoGEFs and two mRNA sequences of rat RhoGAPs that are involved in vasoconstriction. Specific primers for quantitative reverse transcription PCR (qRT-PCR) were designed using Primer3Plus to amplify fragments ranging from 103 to 146 bp (Table 1). The primers were synthesized by Fasmac Co., Ltd. (Kanagawa, Japan).

Total RNA was extracted from the prepared smooth muscle strips using the RNeasy Fibrous Tissue Mini Kit (Qiagen, Hilden, North Rhine-Westphalia, Germany) according to the manufacturer’s instruction. First-strand cDNA was synthesized using the PrimeScript™ RT Master Mix (Takara, Shiga, Japan). The resulting cDNA was combined with TB Green^®®^ Premix Ex Taq™ II (Takara, Shiga, Japan) and the specific primers, and qPCR was performed. PCR amplicons were analyzed on a 2% agarose gel using a 100 bp DNA ladder (Takara, Shiga, Japan) as a size standard.

### 4.4. Protein Extraction and Co-Immunoprecipitation

The prepared smooth muscle strips were stimulated with 60 mM K^+^ for 5 min, with or without prior treatment with 10 mM sodium salicylate for 20 min. After stimulation, the strips were homogenized in RIPA buffer (Wako, Osaka, Japan) containing 1% protease inhibitor cocktail (Sigma-Aldrich, St. Louis, MO, USA), 1% phosphatase inhibitor cocktail 2 (Sigma-Aldrich, St. Louis, MO, USA) and 0.1% Triton^®®^ X-100 (Millipore, Darmstadt, Germany), and then vortexed for 2 h. After vortexing, the sample was centrifuged, and the supernatant was collected as the extracted protein.

Dynabeads™ Protein G (Invitrogen, Waltham, MA, USA) were incubated with either anti-phospho-Pyk2 (Tyr402) antibody (Santa Cruz, Dallas, TX, USA) or normal mouse IgG (Sigma, St. Louis, MO, USA). The extracted protein was added to the antibody-bound Dynabeads™ and incubated for 30 min. Following the washing steps, the beads were resuspended in extraction buffer supplemented with 4× Laemmli sample buffer (Bio-Rad, Hercules, CA, USA) and heated at 95 °C for 10 min. The resulting supernatant was collected and used as the protein sample for subsequent analysis.

### 4.5. Western Blotting

Samples (45 μL) were subjected to SDS-PAGE using 7.5% acrylamide gels. Following electrophoresis, proteins were transferred to PVDF membranes (Bio-Rad, Hercules, CA, USA), which were then blocked with PVDF Blocking Reagent (Toyobo, Osaka, Japan) for 2 h at room temperature. The membranes were subsequently incubated with primary antibody for 16 h at 4 °C, followed by incubation with the secondary antibody for 2 h at room temperature. Protein bands were visualized using enhanced chemiluminescence (ECL) and quantified by densitometry. Actin, which is abundantly expressed in smooth muscle, was used as an internal standard, and the band intensities were normalized to the actin band stained with Coomassie Brilliant Blue (CBB) following SDS-PAGE. The following primary antibodies were used: rabbit polyclonal anti-ArhGEF1 antibody (1:500, Invitrogen, Waltham, MA, USA), rabbit polyclonal anti-ArhGEF2 antibody (1:1000, Invitrogen, Waltham, MA, USA), rabbit polyclonal anti-ArhGEF11 antibody (1:1000, Assay Biotechnology, San Jose, CA, USA), mouse monoclonal anti-ArhGEF12 antibody (1:1600, Santa Cruz, Dallas, Texas, USA), rabbit polyclonal anti-ArhGEF18 antibody (1:500, GeneTex, Irvine, CA, USA), rabbit polyclonal anti-ArhGEF25 antibody (1:2000, Proteintech, Rosemont, IL, USA), mouse monoclonal anti-ArhGAP35 antibody (1:100, Santa Cruz, Dallas, Texas, USA), rabbit polyclonal anti-phospho-ArhGAP42 (Tyr792) antibody (1:1000, MyBioSource, San Diego, CA, USA), rabbit polyclonal anti-phospho-ArhGAP42 (Tyr376) antibody (1:500, Invitrogen, Waltham, MA, USA), and rabbit polyclonal anti-phospho-Pyk2 (Tyr402) antibody (1:1000, Invitrogen, Waltham, MA, USA). The following secondary antibodies were used: goat Anti-Rabbit IgG antibody (Sigma, St. Louis, MO, USA) and mouse-IgGκ binding protein-HRP antibody (Santa Cruz, Dallas, TX, USA), both diluted 1:20,000 dilution.

### 4.6. Statistical Analysis

Data represent the mean  ±  standard error of the mean (S.E.M.). Values of *n* indicate the numbers of rats used in each experiment. Statistical comparisons between two groups were performed using Student’s *t*-test. For comparisons among three or more groups, one-way ANOVA followed by the Tukey–Kramer multiple comparisons test was used. *p*-Values of <0.05 were considered statistically significant. All statistical analyses were conducted using JMP Pro 17 software (SAS Institute Japan, Tokyo, Japan).

## 5. Conclusions

In this study, we reveal a novel Pyk2/ArhGAP42/RhoA signaling pathway whereby membrane depolarization elevates [Ca^2+^]_i_, triggering Pyk2 autophosphorylation in rat caudal arterial smooth muscle. Activated Pyk2 phosphorylates ArhGAP42 at Tyr792, inactivating it and thereby enhancing RhoA activity to drive sustained vascular contraction (Figure 4). This mechanism highlights a key link between calcium signaling and RhoA-mediated contractility. Given its central role in modulating the RhoA/ROCK pathway, Pyk2 emerges as a novel therapeutic target for cardiovascular diseases characterized by hypercontractility, including hypertension. Future studies should clarify the precise functional consequences of Tyr792 phosphorylation on ArhGAP42.

## Figures and Tables

**Figure 1 ijms-26-08676-f001:**
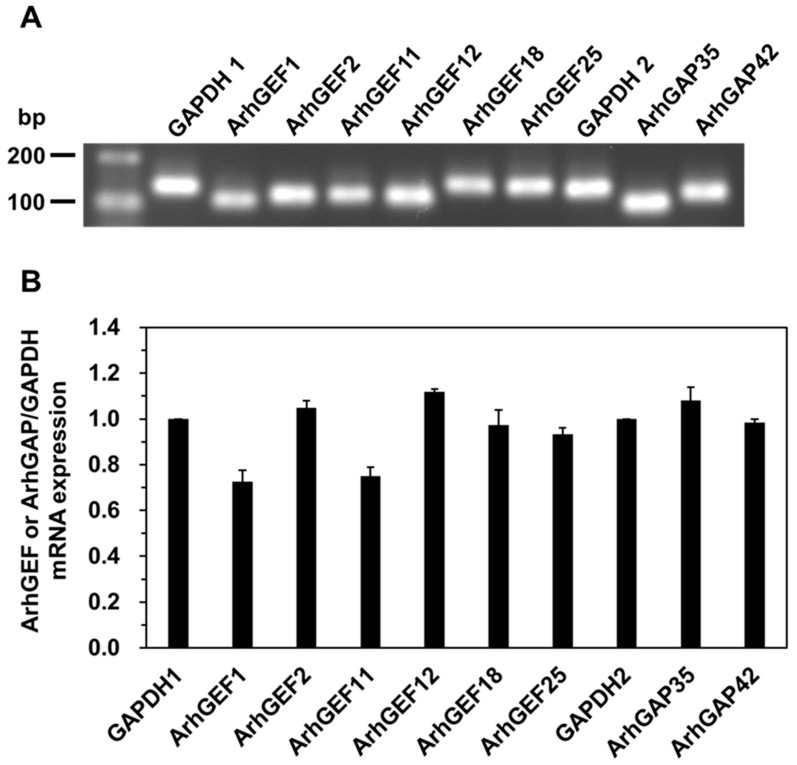
mRNA expression of rat caudal arterial smooth muscle. (**A**) Representative RhoGEFs or RhoGAPs mRNA expression in rat caudal arterial smooth muscle. (**B**) Cumulative mRNA expression of each RhoGEF and RhoGAP was assessed using relative quantification of the PCR products. Values represent the mean ± S.E.M. (*n* = 3).

**Figure 2 ijms-26-08676-f002:**
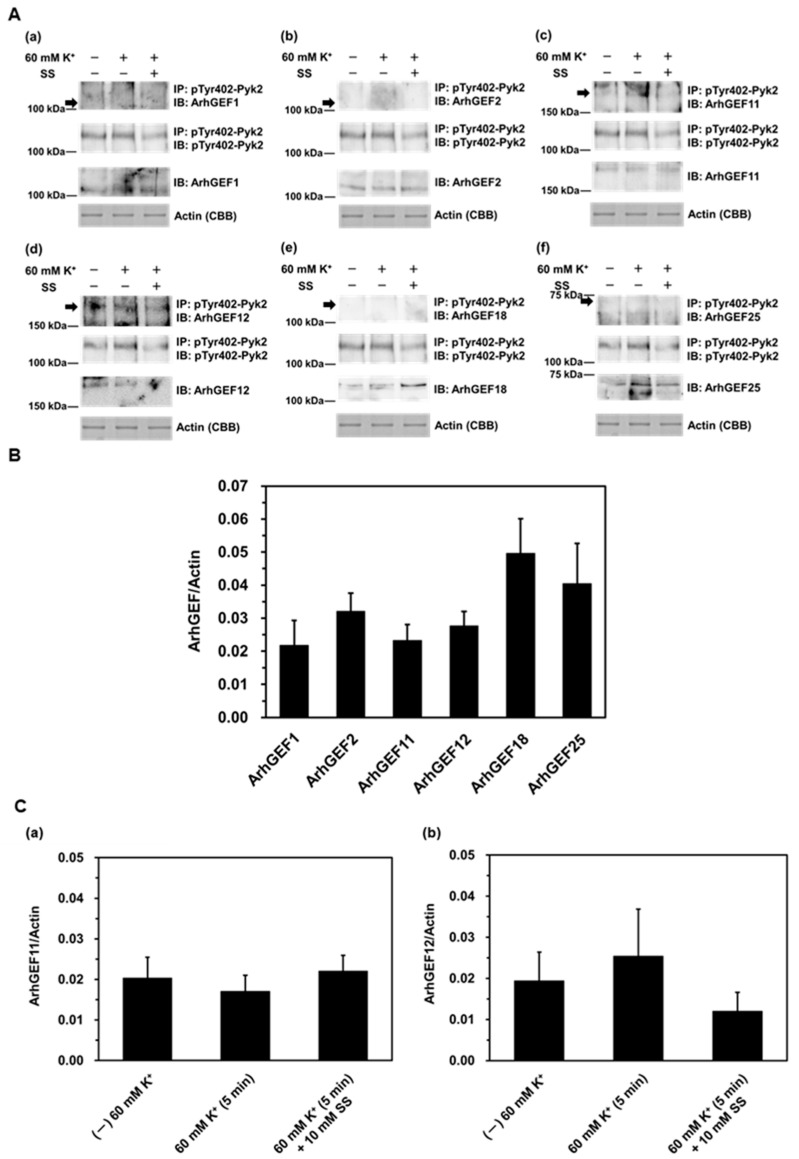
Analysis of relationship between RhoGEFs and phosphorylated Pyk2 using co-immunoprecipitation and Western blotting. (**A**) Representative Western blots showing the co-immunoprecipitation of phosphorylated Pyk2 at Tyr402 (pTyr402-Pyk2) with RhoGEFs induced by 60 mM K^+^ stimulation and CBB-stained actin, in the absence or presence of 10 mM sodium salicylate (SS). Arrows indicate the positions of ArhGEF protein bands as estimated from their molecular weights. (a) ArhGEF1; (b) ArhGEF2; (c) ArhGEF11; (d) ArhGEF12; (e) ArhGEF18; (f) ArhGEF25. (**B**) Cumulative protein levels of each RhoGEF normalized to actin expression in homogenate fractions. Values represent the mean ± S.E.M. (*n* = 4–7). (**C**) Cumulative levels of ArhGEF11 (a) and ArhGEF12 (b) immunoprecipitated with pTyr402-Pyk2 in the absence or presence of 10 mM sodium salicylate, normalized to CBB-stained actin. Values represent the mean ± S.E.M. (*n* = 5). Statistical comparisons of means between groups were performed using one-way ANOVA followed by Tukey–Kramer multiple comparison post hoc test.

**Figure 3 ijms-26-08676-f003:**
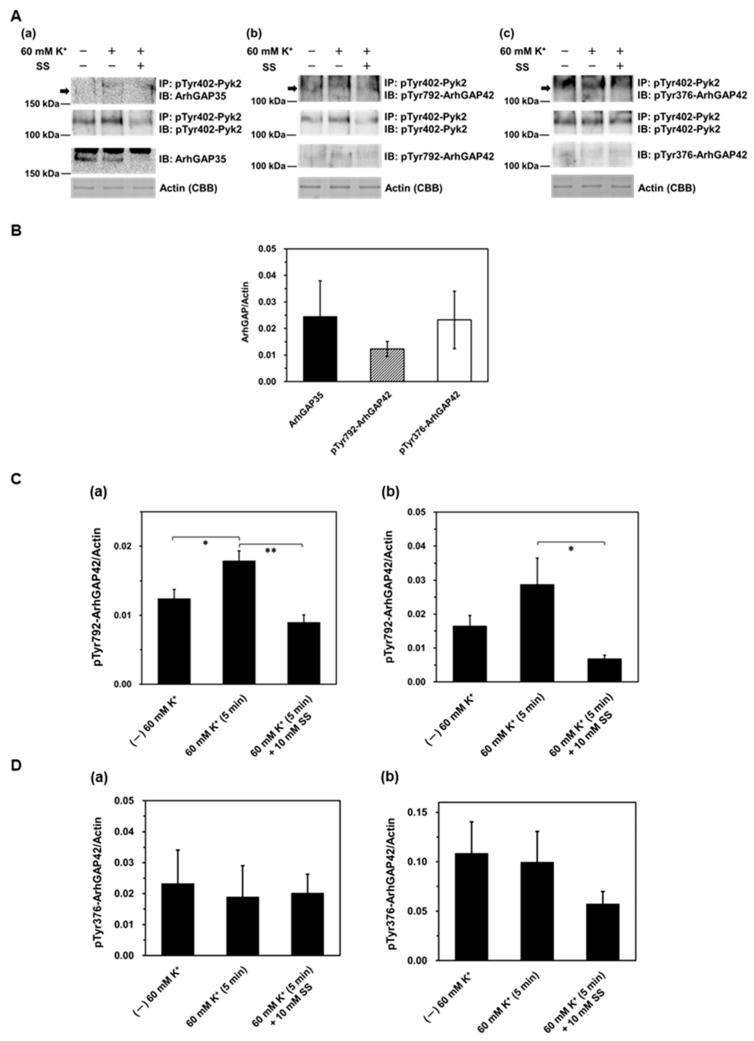
Relationship between RhoGAPs and phosphorylated Pyk2 using co-immunoprecipitation and Western blotting. (**A**) Representative Western blots showing the co-immunoprecipitation of phosphorylated Pyk2 at Tyr402 (pTyr402-Pyk2) with RhoGAPs induced by 60 mM K^+^ stimulation and CBB-stained actin, in the absence or presence of 10 mM sodium salicylate (SS). Arrows indicate the positions of ArhGAP protein bands as estimated from their molecular weights. (a) ArhGAP35; (b) Phosphorylated ArhGAP42 at Tyr792 (pTyr792-ArhGAP42); (c) Phosphorylated ArhGAP42 at Tyr376 (pTyr376-ArhGAP42). (**B**) Cumulative protein levels of each RhoGAP normalized to actin expression in homogenate fractions. Values represent the mean ± S.E.M. (*n* = 4–9). (**C**) Cumulative levels of phosphorylated ArhGAP42 at Tyr792 in homogenate fractions (a) and phosphorylated ArhGAP42 at Tyr792 immunoprecipitated with pTyr402-Pyk2 (b), in the absence or presence of 10 mM sodium salicylate, normalized to CBB-stained actin. Values represent the mean ± S.E.M. (*n* = 4 for (a), *n* = 5 for (b)). (**D**) Cumulative levels of phosphorylated ArhGAP42 at Tyr376 in homogenate fractions (a) and phosphorylated ArhGAP42 at Tyr376 immunoprecipitated with pTyr402-Pyk2 (b), in the absence or presence of 10 mM sodium salicylate, normalized to CBB-stained actin. Values represent the mean ± S.E.M. (*n* = 4 for both (a,b)). Statistical comparisons of means between groups were performed using one-way ANOVA followed by Tukey–Kramer multiple comparison post hoc test (* *p* < 0.05, ** *p* < 0.01).

**Figure 4 ijms-26-08676-f004:**
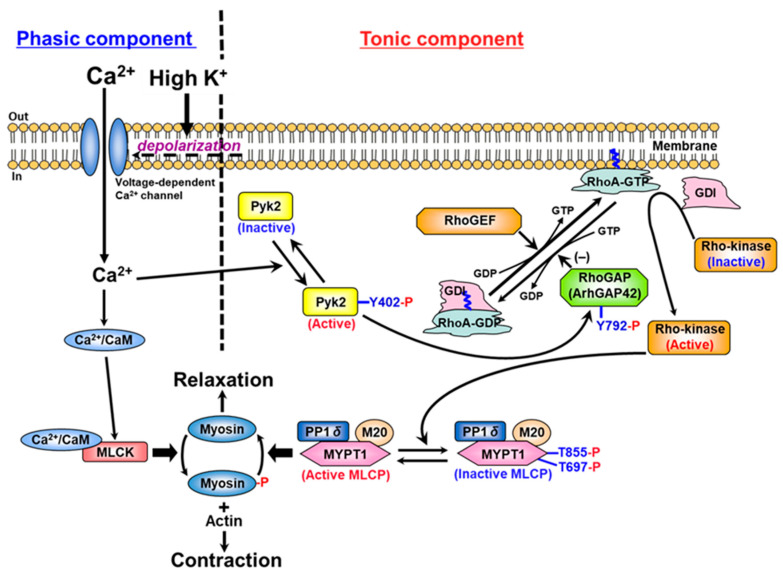
The mechanism of membrane depolarization-induced contraction mediated by Pyk2/RhoA/ROCK in vascular smooth muscle. The addition of 60 mM K⁺ to de-endothelialized strips of rat caudal arterial smooth muscle induced a contraction consisting of two distinct phases: an initial phasic component, which peaked rapidly, and a subsequent tonic component, which declined to a steady level corresponding to approximately 30% of the peak contraction [32]. The underlying mechanisms of these phases can be summarized as follows: (i) the phasic component is characterized by myosin phosphorylation via Ca^2+^/calmodulin-dependent MLCK, and (ii) the tonic component is characterized by MYPT1 phosphorylation through activation of the RhoA/ROCK pathway mediated by Pyk2-dependent inhibition of ArhGAP42. CaM, calmodulin; GDI, GDP dissociation inhibitor; GDP, guanosine 5′-diphosphate; GTP, guanosine 5′-triphosphate; M20, 20 kDa regulatory subunit of MLCP; MLCK, myosin light chain kinase; MLCP, myosin light chain phosphatase; MYPT1, myosin-targeting subunit of MLCP; PP1cδ, a 38 kDa catalytic subunit of type 1 protein phosphatase δ isoform; Pyk2, proline-rich tyrosine kinase 2; RhoGAP, Rho GTPase-activating protein; RhoGEF, Rho guanine nucleotide exchange factor.

**Table 1 ijms-26-08676-t001:** Primer sequences used for qRT-PCR analysis. All sequences are listed in the 5′–3′ direction. GAPDH was used as an internal control.

Gene	Primer	Sequence (5′–3′)	RT-PCR Amplicon
ArhGEF1	ArhGEF1 forward	ATCAAGCTGTCCGTGACATG	109 bp
	ArhGEF1 reverse	TTGAACTCGCTCAGCATTGG	
ArhGEF2	ArhGEF2 forward	AGCATTACAGCCAAGGAAGC	122 bp
	ArhGEF2 reverse	AGCAGTGCAGCTTTCTGTTG	
ArhGEF11	ArhGEF11 forward	TTGTTCAGCGCTGTGTCATC	124 bp
	ArhGEF11 reverse	TTCACACCAGCTTTCATGGC	
ArhGEF12	ArhGEF12 forward	AACAGAAAGTCGAACGCAGCAC	119 bp
	ArhGEF12 reverse	ACAGCGCTGAACAAGACCATAG	
ArhGEF18	ArhGEF18 forward	ATCCGGCAAACTCAAGAACG	146 bp
	ArhGEF18 reverse	GCAAAAGCACATCGGTCAAC	
ArhGEF25	ArhGEF25 forward	TTAAACCGGTGCAGCGAATC	142 bp
	ArhGEF25 reverse	ATATCATTGCAGCGCTTGGG	
ArhGAP35	ArhGAP35 forward	TGCATGCTCTGAAGGAAGTG	103 bp
	ArhGAP35 reverse	ACCTTGTTGTTGTGGCTGAC	
ArhGAP42	ArhGAP42 forward	AGCTGCCTCCAAAAATGTGC	121 bp
	ArhGAP42 reverse	TTGAGCGACCCAGCATTTTC	
GAPDH1	GAPDH1 forward	AGTTCAACGGCACAGTCAAG	119 bp
	GAPDH1 reverse	ATACTCAGCACCAGCATCACC	
GAPDH2	GAPDH2 forward	TGATGCTGGTGCTGAGTATGTC	135 bp
	GAPDH2 reverse	ATCACAAACATGGGGGCATC	

## Data Availability

All the raw data are available from the corresponding author upon reasonable request.

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
