# Peer review of "Role of RhoGEFs or RhoGAPs in Pyk2-Mediated RhoA Activation in Depolarization-Induced Contraction of Rat Caudal Arterial Smooth Muscle"

_ijms, 2025, doi:10.3390/ijms26178676_

Round 1

Reviewer 1 Report

Comments and Suggestions for Authors

Comments to the Author

  The manuscript entitled “Role of RhoGEFs or RhoGAPs in Pyk2-mediated RhoA activation in depolarization-induced contraction of rat caudal arterial smooth muscle” by Aida et al., reports that during membrane depolarization-induced contraction of rat caudal arterial smooth muscle, activated Pyk2 interacts with ArhGAP42 and induces its phosphorylation at Tyr792, which is suppressed by a Pyk2 inhibitor. This study provides a novel observation about the signaling pathway in which Pyk2-mediated phosphorylation of ArhGAP42 contributes to RhoA activation and the sustained contraction of vascular smooth muscle. The work is of potential interest to the readership of IJMS, particularly those investigating smooth muscle contractility and cardiovascular signaling.

  However, several aspects of the study require clarification and additional discussion. In particular, the functional impact of Tyr792 phosphorylation on ArhGAP42 activity remains unresolved, and reliance on antibody-based detection raises concerns about specificity. With additional experimental validation and refinement of the discussion, this work could make a significant contribution to the field.

Major comments

  1. In this study, detection of ArhGAP42 relied solely on a phosphorylation-specific antibody (pTyr792). The authors state that commercially available anti-ArhGAP42 antibodies do not cross-react with the rat protein, and that their own antibody did not bind well, leading to the decision to use anti-phosphorylated (Tyr792) ArhGAP42 antibodies. However, detection of the main band is generally unclear in Western blotting data and the use of anti-phosphorylated (Tyr792) ArhGAP42 antibodies does not allow for accurate assessment of the binding between ArhGAP42 and Pyk2. The limitations of the data should be noted in the discussion section.

  1. The previous report that Y376 phosphorylation of ArhGAP42 leads to GAP activation seems at first glance to be contradictory to the putative inhibitory effect of Y792 phosphorylation in this study. Have you confirmed the effect on the Y376 phosphorylation level of ArhGAP42?

  1. The study demonstrates Tyr792 phosphorylation of ArhGAP42, but its functional consequence on GAP activity remains untested. In vitro GAP assays would substantially strengthen the mechanistic conclusions.

  1. RhoA activation is inferred indirectly. Pull-down assays quantifying RhoA-GTP levels under the tested conditions would provide direct evidence that Pyk2–ArhGAP42 signaling indeed promotes RhoA activation.

  1. Many ArhGEFs have been reported to be involved in RhoA activation during vasoconstriction induced by various physiologically active substances. Although this study demonstrated that ArhGEF11 and 12 associate with PyK2, their functional roles were ruled out. However, their contribution cannot be completely excluded, as it may become evident under different durations of KCl stimulation. Alternatively, Pyk2-mediated phosphorylation of ArhGEF11 might be enhanced, as previously reported in response to angiotensin II stimulation. These considerations should be added to the discussion section.

  1. Sodium salicylate was used as a Pyk2 inhibitor, but it is known to have multiple off-target effects. Employing more potent inhibitors such as PF-431396 would confirm the specificity of the observed effects. Alternatively, if the authors have already shown in their previous studies that these inhibitors have equivalent effects, this should be clearly mentioned in the Discussion.

Minor comments

  1. On Page 5, line2, figure2D is not found in figure2.

  1. On Page 5, line4, Figure2C does not present data regarding ArhGAP.

Reviewer 2 Report

Comments and Suggestions for Authors

This article explores the mechanism by which Pyk2 activates RhoA through the regulation of RhoGEFs or RhoGAPs in depolarization-induced contraction of rat caudal arterial smooth muscle, which holds certain scientific significance. However, the study lacks sufficient novelty and depth in its findings. Although the authors identified the interaction between ArhGAP42 and Pyk2 and its phosphorylation as potentially playing a role in depolarization-induced contraction, this discovery does not significantly advance beyond previous research and fails to clearly elucidate the specific impact of Tyr792 phosphorylation on ArhGAP42 function. Additionally, the co-immunoprecipitation results of ArhGEF11 and ArhGEF12 with Pyk2 lack functional validation, leaving their biological significance unclear. It is recommended to supplement the experiments. Furthermore, the experimental design has flaws. The study uses an anti-phosphorylated ArhGAP42 antibody without validating its specificity, and it does not measure total ArhGAP42 protein levels, limiting the interpretability of the data. The presentation and analysis of the data are also insufficient. Some results lack statistical significance, yet the authors do not discuss the potential reasons or limitations. The discussion section is overly speculative. Therefore, it is recommended to reject the manuscript.

Round 2

Reviewer 1 Report

Comments and Suggestions for Authors

The paper has been appropriately revised in response to the reviewers’ comments. In addition, it would have been desirable to directly assess ArhGAP42 activity or RhoA-GTP levels. Notwithstanding these limitations, the authors have convincingly demonstrated the potentially important role of ArhGAP42 phosphorylation at Tyr792 in Pyk2-mediated RhoA activation during depolarization-induced contraction of rat caudal arterial smooth muscle. On this basis, I recommend publication.

Reviewer 2 Report

Comments and Suggestions for Authors

No other suggestions.